METHODS

# Combined multiplex panel test results are a poor estimate of disease prevalence without adjustment for test error

**Robert Challen**[1,2]*, **Anastasia Chatzilena**[1,2], **George Qian**[1,2], **Glenda Oben**[1,2], **Rachel Kwiatkowska**[3,4], **Catherine Hyams**[1], **Adam Finn**[1], **Krasimira Tsaneva-Atanasova**[5], **Leon Danon**[1,2]

**1** Bristol Vaccine Centre, Schools of Population Health Sciences and of Cellular and Molecular Medicine, University of Bristol, Bristol, United Kingdom, **2** Department of Engineering Mathematics, University of Bristol, Bristol, United Kingdom, **3** Population Health Sciences, University of Bristol, United Kingdom, **4** NIHR Health Protection Unit in Behavioural Science and Evaluation, University of Bristol, United Kingdom, **5** Department of Mathematics and Statistics, University of Exeter, United Kingdom

* rob.challen@bristol.ac.uk

**Data Availability Statement:** All data and code used for running experiments, model fitting, and plotting is available on a GitHub repository at https://bristol-vaccine-centre.github.io/testerror/.

## Abstract

Multiplex panel tests identify many individual pathogens at once, using a set of component tests. In some panels the number of components can be large. If the panel is detecting causative pathogens for a single syndrome or disease then we might estimate the burden of that disease by combining the results of the panel, for example determining the prevalence of pneumococcal pneumonia as caused by many individual pneumococcal serotypes. When we are dealing with multiplex test panels with many components, test error in the individual components of a panel, even when present at very low levels, can cause significant overall error. Uncertainty in the sensitivity and specificity of the individual tests, and statistical fluctuations in the numbers of false positives and false negatives, will cause large uncertainty in the combined estimates of disease prevalence. In many cases this can be a source of significant bias. In this paper we develop a mathematical framework to characterise this issue, we determine expressions for the sensitivity and specificity of panel tests. In this we identify a counter-intuitive relationship between panel test sensitivity and disease prevalence that means panel tests become more sensitive as prevalence increases. We present novel statistical methods that adjust for bias and quantify uncertainty in prevalence estimates from panel tests, and use simulations to test these methods. As multiplex testing becomes more commonly used for screening in routine clinical practice, accumulation of test error due to the combination of large numbers of test results needs to be identified and corrected for.

## Author summary

During analysis of pneumococcal incidence data obtained from serotype specific multiplex urine antigen testing, we identified that despite excellent test sensitivity and specificity, the small error rate in each individual serotype test has the potential to compound and cause large uncertainty in the resulting estimates of pneumococcal prevalence, obtained

This is in the form of an R package providing methods to support the estimation of epidemiological parameters based on the results of multiplex panel tests and it is deployed on the Bristol Vaccine Centre r-universe (https://bristol-vaccine-centre.r-universe.dev/testerror). We have also used Zenodo to assign a DOI to the repository: doi:10.5281/zenodo.7691196.

**Funding:** RC and LD are funded by UK Research and Innovation AI programme of the Engineering and Physical Sciences Research Council (EPSRC grant EP/Y028392/1; https://www.ukri.org/councils/epsrc/). RC and LD are affiliated with the JUNIPER partnership funded by the Medical Research Council (MRC grant MR/X018598/1; https://www.ukri.org/councils/mrc/). KTA gratefully acknowledges the financial support of the Engineering and Physical Sciences Research Council (EPSRC grant EP/T017856/1; https://www.ukri.org/councils/epsrc/). CH is funded by an Academic Clinical Fellowship from the National Institute for Health Research (NIHR grant ACF-2015-25-002; https://www.nihr.ac.uk/). The views expressed are those of the authors. The funders had no role in study design, data collection and analysis, decision to publish, or preparation of the manuscript.

**Competing interests:** I have read the journal's policy and the authors of this manuscript have the following competing interests: CH is Principal Investigator and AF is the Chief Investigator of the AvonCAP study which is an investigator-led University of Bristol study funded by Pfizer https://www.bristol.ac.uk/translational-health-sciences/research/bcvc/research/avoncap-study/. RC, AC, GQ, GO, RK, and LD also receive research funding from Pfizer via the AvonCAP study. AF leads another project investigating transmission of respiratory bacteria in families jointly funded by Pfizer and the Gates Foundation. AF is a member of the Joint Committee on Vaccination and Immunization (JCVI) pneumococcal subcommittee. Funding for the AvonCAP study was provided by Pfizer, however, the manuscript development and the analysis that is the subject of this manuscript were conducted independently of the AvonCAP study and Pfizer.

by combining individual results. This limits the accuracy of estimates of the burden of disease caused by vaccine preventable pneumococcal serotypes, and in certain situations can produce marked bias.

This is a *PLOS Computational Biology* Methods paper.

## Introduction

Multiplex panel testing is a convenient and rapid diagnostic approach and is increasingly being used in clinical practice to differentiate between viral and bacterial causes of a range of disorders [1]. It has also been used in epidemiological studies to identify pneumococcal subtypes targeted by vaccines [2] or monitor disease spread [3]. Multiplex panel tests have been developed for a wide range of clinical syndromes caused by different pathogens, or for specific diseases caused by different subtypes of the same pathogen [1], and may be based on immunological [4, 5] or genetic techniques [6–11]. The number of targets tested for in each multiplex are increasing, but range from a handful, up to 48 different causative agents [3]. In this paper we demonstrate that when large multiplex panels are used, even small errors in the component tests can cause significant compound error and potential bias if the results are combined, usually leading to an overestimate of the prevalence of the combined condition.

In the schematic in Fig 1, we distinguish between multiplex testing (Fig 1A–1D) and other types of multiple testing (Fig 1E–1G). Fig 1A–1D show two component tests which identify each of two subtypes of disease. The disease subtypes are present independently of each other and the disease super-type is present if any of the subtypes is present (Fig 1B and 1C). In Fig 1A we see that a false positive in one component, results in a false positive in the combined panel. In Fig 1B one subtype is correctly detected, in Fig 1C the other subtype, and in Fig 1D a false positive result for one subtype and a false negative for the other results in an overall result which is correct for the wrong reason. In all Fig 1A–1D, the combined test result would be interpreted as positive. As described above, this design of test is usually extended to many more than two subtypes to make a multiplex panel.

Fig 1E–1H show a different test design which is more related to multiple modalities of testing [12]. In this situation, the multiple tests are looking for the same underlying cause of disease which does not have subtypes. In Fig 1E, both tests are true negatives and the overall result also a true negative. The interpretation of the two tests can be: a) that any single test being positive infers disease, in which case all Fig 1F–1H show positive combined results, or b) that both tests must be positive to identify the disease, in which case only Fig 1H represents a positive result. These are not regarded as multiplex tests.

In more formal language, we assume a multiplex panel consists of a set of component tests which test different hypotheses, the results of which are combined to give a panel result where a positive test result in any component implies a positive test result in the panel. It is assumed that subtypes of disease are independent, and a second subtype may co-occur in a patient with one subtype, with the same likelihood. We also assume that test errors are also independent and do not correlate with either disease or other test results.

If a condition is composed of many subtypes, then the prevalence of each individual subtype must be less than or equal to the overall condition prevalence; and on average the subtype prevalence must be less, even when we account for the possibility of co-infection with multiple

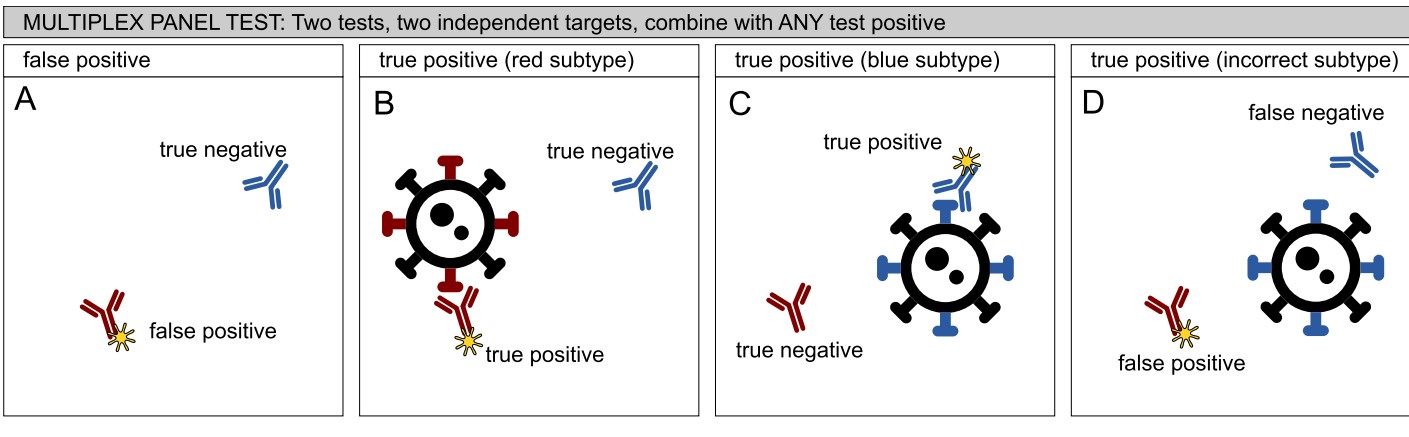

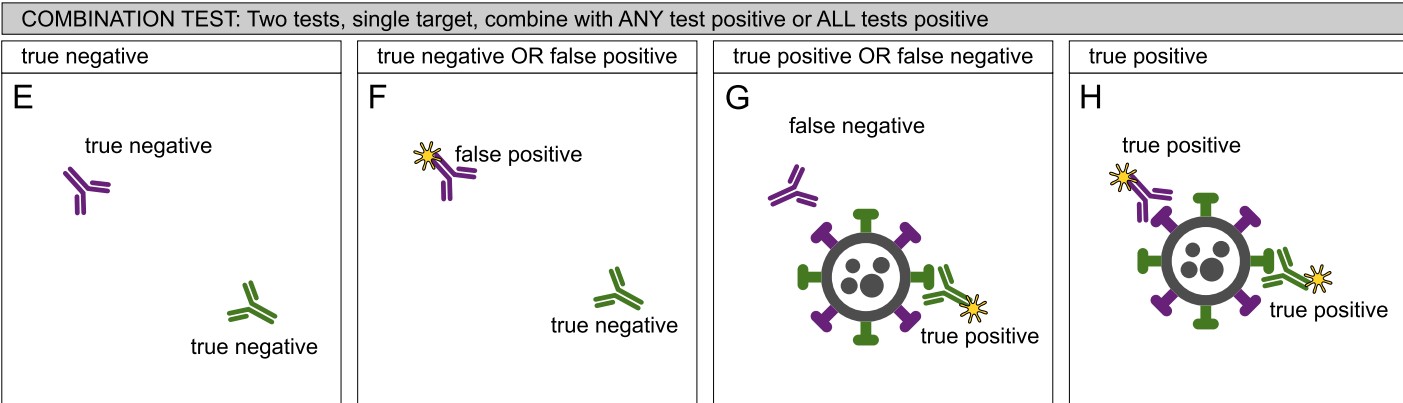

**Fig 1. Two scenarios for multiple testing.** Fig 1A–1D depict a multiplex panel test which is the subject of this analysis. It depicts the situation where multiple tests are employed to detect multiple subtypes of disease which may be present separately or together, the results of which are combined to give an overall result, such that if any component test is positive, the combination is positive. An alternative, shown in Fig 1E–1H, and not in scope of this paper concerns the situation where multiple tests are used to identify a single condition. In this case two interpretations of the multiple test results are possible, which either maximise test sensitivity or test specificity.

subtypes. The more subtypes in a multiplex panel, the smaller that average prevalence will be. If the prevalence of a particular component is low, then that component test is operating at a level where the positive predictive value of the test (i.e. the probability that a positive test result represents a true positive rather than a false positive) is also relatively low. This leads to a high probability of observing false positives in low prevalence components. We will also observe false negatives depending on the sensitivity of the test, but if the prevalence of a subtype is low, there are fewer true positives to be missed.

The effect of this can be seen in Fig 2 where we look at the theoretical distribution of false negatives and false positives in 1000 tests for three hypothetical disease subtypes, present at 2%, 0.5% and 0% prevalence, assuming a test with high specificity of 99.75% and moderate sensitivity of 80%. At 2% prevalence, false positive test results are likely to be balanced by the false negatives (Fig 2A) and the expected test positivity is expected to be lower than 2%, the true value of prevalence in this simulation, (Fig 2B and 2C). When the prevalence of the subtype is lower, at 0.5%, this pattern is reversed, and the false positives will tend to outweigh the false negatives (Fig 2D) leading to a higher test positivity than prevalence (Fig 2E and 2F). In the 0% scenario (Fig 2G, 2H and 2I) all positives are by definition false positives, distributed with high variance leading to a test positivity above 0.

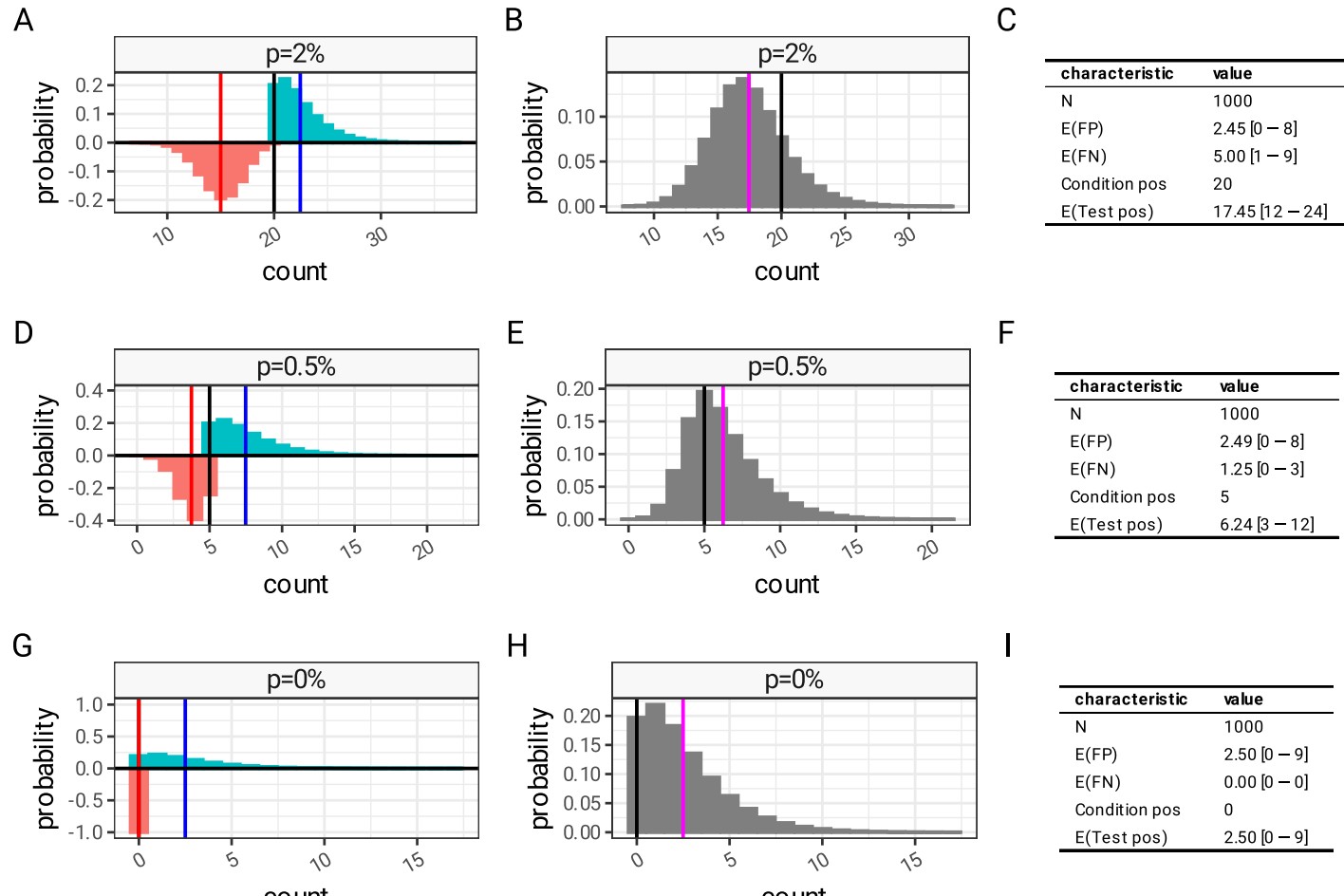

**Fig 2. Error distributions of test results in low pre-test probability settings.** Distribution of false positives (cyan bars, with expected value, $E$(FP), as a blue vertical line) and false negatives (orange bars, expected value, $E$(FN), red line) of 1000 hypothetical test results with 0.9975 specificity and 0.8 sensitivity at different prevalence levels. (A), (D) and (G) show the disaggregated distribution of false positives and false negatives and (B), (E) and (H) show the combined error distribution of test positive observations (grey bars), and expected test positivity (magenta line, $E$(Test pos)) compared to the true condition positives (black line). (C) shows numerically the parameters plotted in (A) and (B); (F) relates to (D) and (E); and (I) to (G) and (H).

If a multiplex panel which consists of 20 subtypes is applied to a disease which is present at a prevalence of 10%, then it is reasonable to expect that the three patterns in Fig 2 will be present in some combination. The components have a mix of false positives and false negatives, in a manner dependent on the distribution of disease subtypes. In this particular scenario (20 highly specific tests at 10% prevalence) the balance of these will be towards false positives. Because any positive component results in a positive panel result, the component false positive errors compound in combination. In this example the error combines in such a way that the panel result will contain more false positives than false negatives, and the resulting test positivity rate will be an overestimate of true prevalence.

The compounding of error in numerous components is analogous to parallel testing of multiple statistical hypotheses. In this situation, a Bonferroni correction is often used to reduce the risk of over-interpreting the results of statistical tests of significance [13]. In a similar way, results from parallel testing of disease sub-types are at risk of being over-interpreted without a clear understanding of the nature of test errors.

In the remainder of this paper we quantify this risk, and summarise the mathematical properties of multiplex tests, under the assumption of independence of the subtype disease, meaning that any individual patient may be co-infected with multiple disease subtypes at once, and these do not compete with each other. This behaviour is common in diseases caused by commensal organisms, or in disease syndromes caused by multiple pathogens.

We use a realistic simulation based on the example of pneumococcal serotypes to demonstrate the implications and study potential mitigation strategies. In S1 Appendix we provide the detail of the mathematical analysis, and validate our findings against a broad range of simulation scenarios. In S2 Appendix we provide specific detail on propagation of uncertainty associated with combined multiplex panel testing, and validate this against a set of realistic simulations. Supporting implementations of all methods described here are provided in the associated R package "testerror" (https://bristol-vaccine-centre.github.io/testerror/).

## Materials and methods

In this section we describe the mathematical analysis, the methods used to adjust for potential bias and uncertainty, and the simulations used to test and illustrate the problem. The majority of the detailed methods are found in S1 and S2 Appendices. The equations presented here are for ease of reference and are not essential to the remainder of the analysis presented in this summary paper.

### Mathematical analysis and validation

Given a set of $N$ multiplex panel component tests, the combined test result is defined as positive if any of the panel component tests are positive. It follows that a true negative panel result can only be the result of a combination of all true negative components. In S1 Appendix we use this to determine expressions for estimates of sensitivity ($sens_N$) and specificity ($spec_N$) of combined panels as shown below. In Eqs 1 and 2, $\widehat{AP_n}$ is the apparent prevalence (test positivity rate) for the component tests, and $\widehat{AP_N}$ for the combined panel. $sens_n$ is the sensitivities of the component tests with $spec_n$ as the component specificities.

$$spec_N = \prod_{n \in N} spec_n$$

$$\widehat{sens_N} \approx 1 - \frac{\prod_{n \in N}(1 - \widehat{AP_n}) - \prod_{n \in N} spec_n \times \frac{sens_n - \widehat{AP_n}}{spec_n + sens_n - 1}}{1 - \prod_{n \in N} \frac{sens_n - \widehat{AP_n}}{spec_n + sens_n - 1}} \quad (1)$$

There is the possibility that a false negative test result in one component can be "corrected" by a false positive result in another component, resulting in a panel result that is correct but for the wrong reason, or otherwise masked by a true positive detection of another co-occurring component. This leads to the panel sensitivity being a complex expression which counter-intuitively depends on the prevalence of the condition it is measuring.

From these expressions for panel sensitivity and specificity, we use the Rogan-Gladen estimator of true prevalence [14], to derive an estimate for the true prevalence of a combined panel ($\widehat{prev_N}$) based on the test positivity, sensitivity and specificity of the components, and

test positivity of the panel.

$$\widehat{prev_N} \approx \frac{\prod_{n \in N} spec_n - (1 - \widehat{AP_N})}{\prod_{n \in N} spec_n - \prod_{n \in N} (1 - \widehat{AP_n})}\left(1 - \prod_{n \in N} \frac{sens_n - \widehat{AP_n}}{spec_n + sens_n - 1}\right) \qquad (2)$$

In S1 Appendix these estimators are demonstrated to perform well in a broad range of scenarios based on randomly generated synthetic multiplex panels, and the behaviour of these estimators is analysed in detail.

## Application to realistic situations

To illustrate the implications of multiplex test error for epidemiological studies, we have constructed a simulation based on pneumococcal serotypes. There are over 100 pneumococcal serotypes, and mixed serotype carriage is common, particularly in children. Commensal pneumococcal serotypes opportunistically cause disease, and during disease episodes, multiple serotypes are occasionally detected when tested for. For this simulation we focus on the 20 serotypes that are covered by the 20-valent polysaccharide capsule vaccine (PCV20).

We previously published the frequency of the pneumococcal serotypes contained in PCV20, that were identified in an invasive pneumococcal disease (IPD) cohort in Bristol between January 2021 and December 2022 [15]. This IPD distribution was scaled to give a realistic distribution of 20 subtypes in a hypothetical population with an overall PCV20-type pneumococcal prevalence of 10%. We simulate testing this population with a hypothetical multiplex panel which detects the 20 individual serotypes. For illustration purposes, we assume all component tests of the multiplex panel are moderately sensitive (80%) and highly specific (99.75%), (these assumptions are loosely based on existing serotype specific detection tests). The simulated test results for individual serotypes were aggregated into a PCV7 group (any positive of serotypes 4, 6B, 9V, 14, 18C, 19F, 23F), a PCV13 group (PCV7 groups plus 1, 3, 5, 6A, 7F, 19A), a PCV15 group (PCV13 plus 22F and 33F), and a PCV20 group (all serotypes). This allows us to compare "true" simulation prevalence to test positivity rates (apparent prevalence). Using the estimators for panel sensitivity and specificity above, we use the synthetic data set to estimate the true prevalence from test positivity, of both components and panels. With the same basic simulation we vary component test sensitivity and specificity, and investigate how the difference between "true" simulation prevalence (10%) and simulated test positivity rates (apparent prevalence) depends on test performance in a realistic scenario.

## Uncertainty propagation

Our mathematical analysis assumes precisely known values for the specificity and sensitivity of component tests. However, these quantities can only be estimated as a result of control-group testing. Because individual subtypes are usually present at low levels when there are multiple subtypes, the number of positive disease controls for any given subtype is typically small [2]. This places a limit on the precision of estimates of component test sensitivity, which in turn makes interpretation of test positivity in both components and panels challenging.

For single tests, there are approaches to estimating true prevalence from test positivity, which incorporate uncertainty in sensitivity and specificity, in both frequentist [16–18] and Bayesian frameworks [18–20]. In S2 Appendix we extend these two frameworks to account for multiplex testing, and implement a third resampling procedure combined with the Rogan-Gladen estimator to propagate uncertainty. We test this against a synthetic data set that is based on the IPD distribution scaled to an overall pneumococcal prevalence of 10% (further

described in S2 Appendix). The methods described here are provided in the associated R package "testerror" (https://bristol-vaccine-centre.github.io/testerror/).

## Results

In the illustrative simulation motivated by IPD serotype distributions, the serotypes range from having no observed cases to making up 25.6% of the total [15]. When this is scaled to a synthetic population with 10% overall prevalence, the component prevalence ranges from 0% to 3.8% and, as with the theoretical examples in Fig 2D–2I, the majority of serotypes fall into the category where the apparent prevalence is higher than the true prevalence due to false positives, despite assuming a highly specific test with 99.75% specificity (Fig 3A). The bias towards overestimation due to false positives is strongest for subtypes with low, or zero, prevalence, whereas the underestimation due to low sensitivity is strongest for subtypes with higher prevalence (also demonstrated in Fig 2A–2C).

In the synthetic but realistic scenario in Fig 3A, with excellent test specificity (99.75%) and moderate test sensitivity (80%), test positivity rate (apparent prevalence) is expected to be higher than true prevalence under a threshold of 1.2%. When a set of 20 components are combined, that together result in a true panel prevalence of 10%, the combined errors mean that the panel test positivity is higher than the true prevalence (Fig 3B, dashed black lines). In Fig 1D and S1 Appendix we identify that false positives in one test balance out false positives in another test, and this makes panel test sensitivity a complex quantity that counter-intuitively depends on disease prevalence, component distribution, sensitivity and specificity. As a result, the relationship between true panel prevalence and apparent panel prevalence (test positivity) is non-linear (Fig 3B), and in this particular simulation, test positivity will be an over-estimate of true prevalence, until true prevalence exceeds 22%.

Component sensitivity and specificity determine the difference between true and apparent prevalence as shown in Fig 4. This considers the same scenario of 10% prevalence, but shows

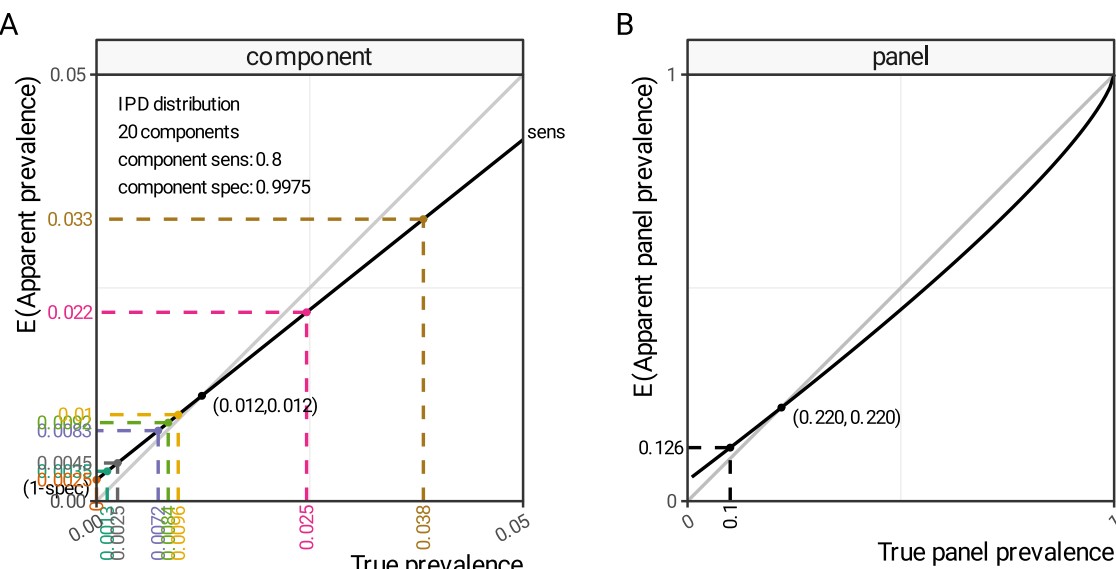

**Fig 3. True versus apparent prevalence in multiplex test components and panel results.** The apparent prevalence as a function of true prevalence in a simulated realistic scenario with excellent test specificity (99.75%) and moderate test sensitivity (80%). Fig 3A shows the individual component relationship and Fig 3B shows the panel relationship when 20 components are combined. Black lines show the relationship and the grey transparent lines are a guide to the eye showing perfect agreement. Note that Fig 3A and 3B are on very different scales.

the relative difference between true and apparent prevalence when varying sensitivity and specificity. The previous assumptions are marked as a blue cross in the figure, and at this high level of specificity (i.e. 99.75%—right dotted vertical line in Fig 4) the ratio between apparent and true prevalence is mostly influenced by test sensitivity. If sensitivity is low enough (less than 50%) the false negative rate exceeds the combined false positive rate and apparent prevalence is smaller than true prevalence. In any situation where the specificity is lower, the balance of error is most influenced by test specificity, and test sensitivity becomes much less important as a factor determining the difference between true and apparent prevalence. Even marginally lower values of test specificity result in test positivity being a gross overestimate of panel prevalence. If the component test specificity is only 98% (left dotted line) the combined 2% false positive rate of 20 components is sufficient to drive the overall panel test positivity to 4 times the level of the true prevalence set in this simulation, regardless of the test sensitivity.

We have described that even low false positive rates in component tests lead to overestimates of uncommon components. The converse is true for components with comparatively high prevalence. In the scenario we have been using as an example, despite the excellent specificity of the tests and 10% overall prevalence the balance of the component estimates is such that test positivity will overestimate true prevalence. This is seen more clearly in Fig 5 (left subfigure) in which simulated true prevalence levels (blue) are lower than test positivity (red) for all but two of the components (serotypes 3 and 8). In the right subfigure we see the effect of

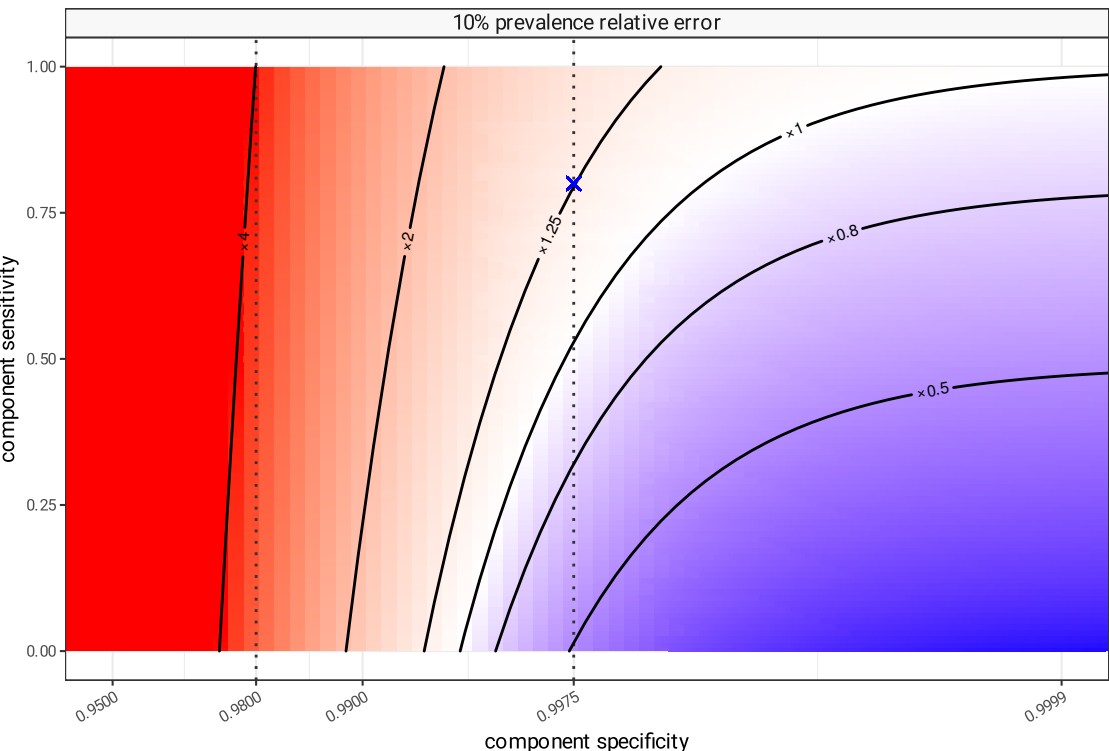

**Fig 4. Bias in apparent prevalence as an estimator for true prevalence.** A simulated scenario of 20 components realistically distributed following patterns seen in IPD, with a simulated true prevalence of 10%, and assuming the same sensitivity and specificity for each of the component tests. Expected test positivity rates are calculated for all combinations of sensitivity and specificity, and compared to the true prevalence (10%) as a ratio. At sensitivity of 80% and specificity of 99.75% (the blue cross) the test positivity rate will be about 1.26 times higher than true prevalence. Blue areas represent parameter space where test positivity is an underestimate of true prevalence due to excess of false negatives, and red areas where test positivity is an overestimate due to excess of false positives. The specificity, presented here on the x-axis, is logit scaled.

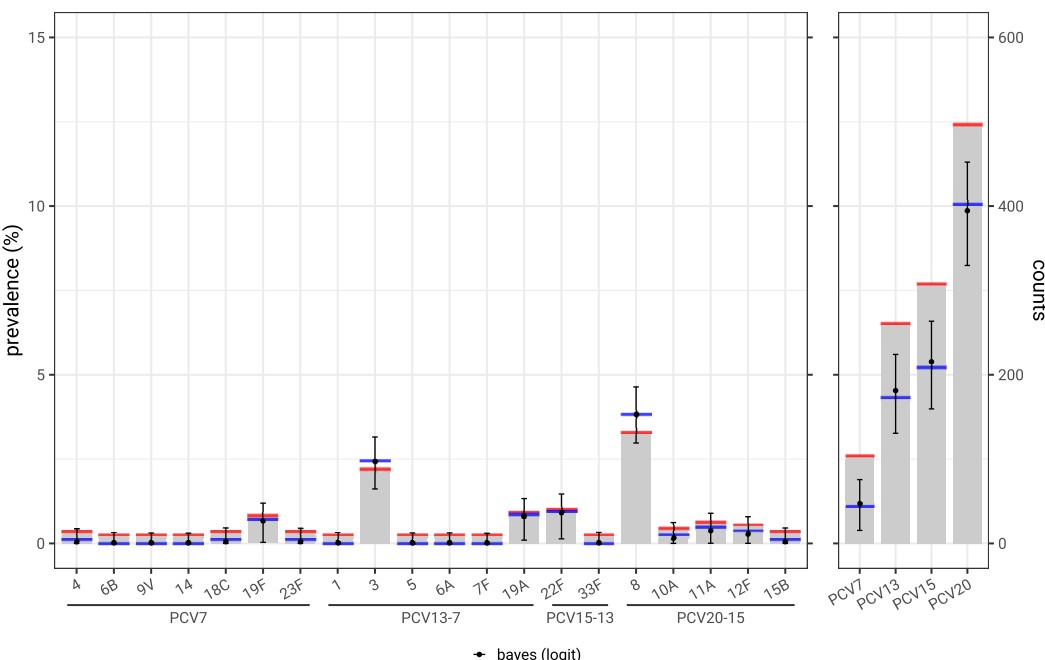

simulated component test performance: sensitivity 80.00%, specificity 99.75%
simulated panel prevalence: PCV7 − 1.10%; PCV13 − 4.34%; PCV15 − 5.26%; PCV20 − 10.00%

**Fig 5. Correction of bias in a single IPD scenario.** The relative frequency of the 20 pneumococcal serotypes contained in PCV20, and identified in Bristol within the last 2 years, informed a simulation of a serotype distribution with an overall PCV20 pneumococcal prevalence of 10% (blue lines) in a sample size of 4000 synthetic patients. Test positivity was simulated assuming each serotype test had a sensitivity of 80% and a specificity of 99.75% (red lines) resulting in underestimates of 'true' prevalence for serotypes 3 and 8, and overestimates for the rest. In the right subfigure combined test positivity for each PCV group (red lines) overestimate true prevalence (blue lines) for this scenario. We estimate true prevalence from test positivity (red lines), incorporating uncertainty in component sensitivity and specificity using a Bayesian model described in S2 Appendix. These estimates are shown as point estimates and 95% credible intervals (black), which accurately estimate the true prevalence.

combining these into groups of 7, 13, 15 and 20 components, representing combinations of serotypes targeted by vaccines. As predicted, overestimates of prevalence are compounded and the size of each overestimate depends both on the number and distribution of test components.

In S2 Appendix we describe methods for correcting this bias in both frequentist and Bayesian frameworks using results from the mathematical analysis (S1 Appendix). In Fig 5 the Bayesian correction is applied and we are able to predict the true prevalence (blue) allowing for uncertainty in our knowledge of test sensitivity and specificity. This is examined in a broader range of scenarios in S2 Appendix. The Bayesian analysis assumes an uninformative prior on the prevalence of components and panel, but test results themselves are uninformative about sensitivity and specificity. On the other hand, test results are highly informative about prevalence given knowledge of the sensitivity and specificity. The Bayesian analysis therefore requires informed priors for sensitivity and specificity, and apart from excluding logically impossible values, does not have enough information to also determine sensitivity and specificity. Disease free control group data informs specificity, which is relatively easy to obtain experimentally. Disease positive control group data can be used to inform sensitivity but is practically harder to obtain, particularly at a component level. For frequentist approaches control group data can be used to construct empirical sensitivity and specificity estimates.

In summary both Bayesian and Lang-Reiczigel (frequentist) approaches work well when we have good prior knowledge about test sensitivity and specificity, but if these assumptions are very wrong, then we cannot expect either method to produce accurate estimates.

## Discussion

Combining multiplex test results into a panel commonly results in test positivity that significantly overestimates true prevalence. Multiplex testing simultaneously tests many hypotheses, and by combining the result into a single panel result leads to compounding of error. This error can be significant because of the low positive predictive value of individual component tests operating at low pre-test probability. This is critically dependent on component test specificity, and very high specificity is essential in tests which are designed to be interpreted as a combined result.

Panel test sensitivity is difficult to characterise. When multiplex tests are combined, components with a larger pre-test probability will generate more false negatives. In panel tests, false negative results in one component are over-ridden by any positives in other components. The sensitivity of the overall panel test is therefore a complicated function of component test sensitivity, specificity and pre-test probability (component prevalence), leading to higher panel sensitivity at higher prevalence. This is counter-intuitive as test sensitivity is usually regarded as independent of prevalence. This makes it challenging to compare panel test positivity rates in populations with different prevalence.

Despite the complexities around panel test specificity and sensitivity, it remains possible to estimate true prevalence from test positivity. The count of positive panel tests in a sample is not a binomially distributed quantity, due to false positive and false negative results in the components (see Fig 2). We cannot use binomial confidence intervals for estimates of panel test positivity. Again due to the false positives and false negatives, the raw panel test positivity is a biased estimator of the true prevalence, which commonly overestimates prevalence (see Fig 4). Because of the relationship between panel sensitivity and prevalence, the degree of bias depends on the prevalence we are trying to measure. This makes panel test positivity very hard to interpret, even when the sensitivities and specificities of component tests are unchanged. Panel test positivity from two experiments conducted using identical tests in populations with different true prevalence cannot be compared, nor can any degree of statistical significance be attributed to differences between the populations. To be comparable, we recommend that the results of panel tests are presented as modelled true prevalence estimates, with confidence limits expressing uncertainty arising from measurement error, using the techniques described in this paper.

Sensitivity and specificity assumptions that incorporate uncertainty are critical in producing accurate modelled true prevalence estimates, or drawing conclusions for comparisons of two groups. Specificity estimates for multiplex testing usually rely on a disease free control group, which may also be used to determine cut points to achieve set specificity levels, and can usually give us a reasonable estimate of component test specificity. Determining the sensitivity of the components of a multiplex test is much harder as it needs proven cases of disease with known subtype. These are difficult to find for rare disease subtypes, and gold standard identification of disease subtypes is not always available, or free from error [21, 22]. This results in a great deal of uncertainty in estimates of component test sensitivity. In some situations panel test sensitivity is estimated directly, however as we saw above, panel test sensitivity is dependent on a range of factors including overall prevalence, and component distribution. Any direct estimates of panel sensitivity are not generalisable outside of the specific population tested. The methods presented here for modelling true prevalence from multiplex tests do

allow for the uncertainty in sensitivity and specificity to be propagated appropriately. The accuracy of this correction, however, is dependent on the quality of the estimates of specificity and sensitivity (see S2 Appendix), and systematic bias in either quantity prevents correct estimation of true prevalence. To improve accuracy and narrow the confidence intervals of estimates of prevalence it is far more important to characterise the sensitivity and specificity of the test than increase the sample size of testing. With a poorly understood test it is hard to draw any conclusions from the results of a multiplex panel test; however, the methods presented here remain valuable as a way of detecting when multiplex panel tests are underpowered, as insufficient characterisation of component sensitivity and specificity are exposed in the confidence intervals of modelled true prevalence estimates.

The bias in panel test positivity is an inevitable consequence of combining multiple tests in environments with moderate to low prevalence. It can be mitigated in a number of ways: a) the specificity of the component tests is increased, b) second line confirmatory testing is performed, c) the multiplex test can only be applied to populations with a very high overall disease prevalence. In the last case we may be able to use a multiplex test to determine which subtype of disease is causative if we already know the patient has the disease by using a different test, or using specific clinical diagnostic criteria that select patients with high probability of disease.

This analysis assumes independence of the subtype diseases, and that multiple subtypes of disease may be present in the same individual at once. This may not always be the case, for example, in-host competition between disease subtypes is known to occur in dengue infections in mosquitos [23], and facilitative co-operation between pathogens is also described [24]. Addressing these kinds of diseases would require a different statistical analysis, and we anticipate panel prevalence estimates that assume independence such as those presented here, will be lower than ones which assume competition, although this will be less obvious at low prevalence. We also make the further assumption that test error is independent. Test errors could become correlated if there is close overlap between the epitope being tested for in one subtype versus another, and this is observed in pneumococcal urine antigen detection (UAD) testing [2]. In this scenario we expect the major problem to be false positives associated with true positives in a closely related subtype, which could result in overestimates of the prevalence of the offending subtype. However, the combination of a false positive and true positive in a panel is always correctly interpreted as a true positive, so panel prevalence estimates will be less affected. This is a limitation of the method as it stands at the moment and further research is required to characterise this, but we note that data to support this are scarce.

There are analogous situations where multiplex panel tests are used with similar potential risks. For example the Biofire FilmArray respiratory panel 2.1 is one of a number of multiplex panels directed at respiratory pathogens [1]. It detects 19 viruses [21, 25]. We have trialled using this in Bristol to investigate co-infection of respiratory pathogens. There are multiple comparative evaluations of the Biofire FilmArray panel [7, 21, 22, 26–28] but there has not yet been a large scale evaluation of test specificity using disease free controls for each individual panel. Identifying a patient as having co-infection by any of the 19 viral diseases in the panel, requires similar adjustment for the combined test uncertainty of all of the panel components to estimate co-infection frequency.

## Conclusion

In this paper we have characterised the degree of uncertainty that results if multiplex panel test results are combined to give an overall result. The principal example of this is pneumococcal disease, in which specific component tests of a urine antigen detection test (UAD) identify up to 24 individual pneumococcal serotypes [2, 4]. This is designed to be highly specific with

individual serotype tests being around 99.75%. The serotypes are generally grouped together by the vaccines that target them, to determine vaccine preventable disease, or all together as an estimate of pneumococcal disease burden [15]. This use of multiplex UAD testing is susceptible to the uncertainty and biases described in this analysis. Even considering the highly specific nature of the UAD tests [4], as the number of components increases so does the risk of bias. Any seemingly minor decrease in test specificity is expected to have a large impact on estimates of disease burden. Despite excellent specificity, without correction, the large number of tests in the panel creates uncertainty in prevalence estimates using UAD tests, and difficulty in comparing results to those of other similar studies. In this analysis we present methods to correct and quantify uncertainty in prevalence estimates using multiplex panels such as the UAD. These methods are a useful tool but critically rely on estimates of test sensitivity and specificity, and without these it is very hard to estimate disease burden using UAD results.

Uncertainty in test results due to lower sensitivity and specificity result in more noise at lower levels of prevalence [29, 30]. In vaccine effectiveness studies using a test negative design this phenomenon acts to mask the effect of a vaccine in the lower prevalence vaccinated group. Hence test error always results in an underestimate of vaccine effectiveness [30]. The less sensitive the test, the greater this underestimate. For pneumococcal vaccination, the serotype of pneumococcal disease is determined using urine antigen detection (UAD) test panels [2, 4]. Theory suggests that, because of the issues identified here, conclusions on vaccine effectiveness based on the UAD tests are an underestimate [30]. The underestimate of vaccine effectiveness helps mitigate any bias resulting from test error in disease burden estimates, and hence the anticipated impact of a vaccine in the real world may be relatively unaffected. Further work would be needed to formally assess this.

## Supporting information

**S1 Appendix. Sensitivity and specificity of combined panel tests.** Derivation of the performance metrics and true prevalence adjustments for combination tests.
(PDF)

**S2 Appendix. Propagation of uncertainty of combined panel tests.** Bayesian and frequentist approaches to estimating the uncertainty of panel test results.
(PDF)

## Author Contributions

**Conceptualization:** Robert Challen, Leon Danon.

**Formal analysis:** Robert Challen, Leon Danon.

**Methodology:** Robert Challen.

**Software:** Robert Challen.

**Supervision:** Adam Finn, Krasimira Tsaneva-Atanasova, Leon Danon.

**Validation:** Anastasia Chatzilena, Krasimira Tsaneva-Atanasova, Leon Danon.

**Visualization:** Robert Challen.

**Writing – original draft:** Robert Challen.

**Writing – review & editing:** Robert Challen, Anastasia Chatzilena, George Qian, Glenda Oben, Rachel Kwiatkowska, Catherine Hyams, Adam Finn, Krasimira Tsaneva-Atanasova, Leon Danon.

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
