## [Decision Letter · Decision Letter 0]

6 Feb 2024

Dear Dr Challen,

Thank you very much for submitting your manuscript "Combined multiplex panel test results are a poor estimate of disease prevalence without adjustment for test error." for consideration at PLOS Computational Biology.

As with all papers reviewed by the journal, your manuscript was reviewed by members of the editorial board and by several independent reviewers. In light of the reviews (below this email), we would like to invite the resubmission of a significantly-revised version that takes into account the reviewers' comments.

The manuscript has been read be two experts in the field, and more briefly by myself. We all find the manuscript novel and interesting. Still there are some issues to be taken care of before potentially being accepted for publication in PLoS Comp Bio. Please address the good points raised by both referees. In particular you should include your novel finding in the abstract (ref 1) and address the two main concerns of ref 2.

Kind regards, Tom Britton

We cannot make any decision about publication until we have seen the revised manuscript and your response to the reviewers' comments. Your revised manuscript is also likely to be sent to reviewers for further evaluation.

Sincerely,

Tom Britton

Academic Editor

PLOS Computational Biology

Virginia Pitzer

Section Editor

PLOS Computational Biology

The manuscript has been read be two experts in the field, and more briefly by myself. We all find the manuscript novel and interesting. Still there are some issues to be taken care of before potentially being accepted for publication in PLoS Comp Bio. Please address the good points raised by both referees. In particular you should include your novel finding in the abstract (ref 1) and address the two main concerns of ref 2.

Kind regards, Tom Britton

Reviewer's Responses to Questions

**Comments to the Authors:**

Reviewer #1: General

This paper is a welcome addition to the literature on diagnostic testing. To the best of my knowledge, it is the first to address the context of multiplex testing when 1 or more out of n tests positive equates to a positive diagnosis. The provision of an R package is a bonus,

I felt that the presentation in the paper itself could be streamlined, given that the Supplementary Materials contain everything necessary for anyone who wants to dig into the details of the method. To this reader, the killer diagram is Figure 4, which very nicely shows the counter-intuitive nature, and hence the practical importance, of the solution to the problem.

Specific

Abstract. People who only read the abstract will miss the counter-intuitivity – please add a sentence or two

Line 27. There’s a LaTeX glitch here … “Figure /reffig1 E,”

Lines 33-34. “we define a multiplex test as consisting of a set of independent components which test different independent hypotheses” This is not so much a definition as an assumption (a word you use later) … and it’s critical to everything that follows. It does not help that the word independent is being used in two different senses – independence of results given the true disease state and independence of sub-disease states. I’m not a biologist, but I can imagine circumstances in which test results might not be independent, in either sense. An example would be Kato-Katz tests for the three main species of soil-transmitted helminths. Please call this an assumption and add a sentence or two to the discussion.

Mathematical analysis and validation section. I think some readers might be unnecessarily put off by your notation, which would be a shame. You acknowledge (but in the previous section) that the equations are “not essential to the remainder of the analysis presented in this summary paper” I understand why you need this level of formality in the Supplementary Materials but here I would be inclined to just give verbal statements of the quantities being defined, or at least try to avoid indicator functions and compound subscripts.

Lines 136-137 and S2 “This places a limit on the precision of estimates of component test sensitivity, which in turn makes interpretation of test positivity in both components and panels challenging.” Indeed it does, although not (IMHO) for the reason you imply. Probably the best (and certainly the most elegant) way to account for uncertainty in Se and Sp is through a Bayesian analysis, which I would make a bit more prominent in the paper itself. The point does need to be made explicitly somewhere that apart from excluding logically impossible values, the actual test results are uninformative about Se and Sp, whereas they are highly informative about prevalence given Se and Sp. For many pragmatic Bayesians, this justifies using an uninformative prior for prevalence, but raises disquiet about reliance on necessarly informative priors for Se and Sp. However, you assume that you do have control data available (albeit not as much as you would like) so you can these to construct empirical priors. Granted this gives you total of 2n+1 priors, but it does seem defensible to treat these as independent and convert the known Se and Sp result to a Bayesian posterior by numerical integration. Presenting the Bayesian version in this way might be more transparent to the reader than it appeared (to me at least) in S1 and S2.

Lines 219-220. “This is counter-intuitive as test sensitivity is usually regarded as independent of prevalence.” This important message needs to be given more prominence in the abstract and in the main body of the paper.

Lines 225-226. “the expected value of test positivity is not a binomially distributed quantity” Something wrong with the wording here. An expected value doesn’t follow any probability distribution, it’s a function of the parameters of the distribution with respect to which it is calculated. Do you mean empirical prevalence? I guess not, because if patients give independent results then the number of positive results (however defined) does follow a binomial distribution. So what do you mean?

Reviewer #2: Review is uploaded as an attachment

**Have the authors made all data and (if applicable) computational code underlying the findings in their manuscript fully available?**

Reviewer #1: Yes

Reviewer #2: Yes

PLOS authors have the option to publish the peer review history of their article (what does this mean?). If published, this will include your full peer review and any attached files.

Reviewer #1: No

Reviewer #2: No
---

## [Decision Letter · Decision Letter 1]

8 Apr 2024

Dear Dr Challen,

We are pleased to inform you that your manuscript 'Combined multiplex panel test results are a poor estimate of disease prevalence without adjustment for test error.' has been provisionally accepted for publication in PLOS Computational Biology.

Best regards,

Tom Britton

Academic Editor

PLOS Computational Biology

Virginia Pitzer

Section Editor

PLOS Computational Biology

Associate editor

Both reviewers are happy with the revision and so am I. I hence recommend accepting the manuscript.

Kind regards, Tom Britton

Reviewer's Responses to Questions

**Comments to the Authors:**

Reviewer #1: A very nice paper with important practical implications.

Reviewer #2: I am fully satisfied with the changes the authors made and recommend their decision to restructure the supplementary file S1, re-run the respective simulations and update the plots. All my concerns have been very adequately addressed. The authors have convinced me that --as they put it in their response to my comments-- my "concerns relate[d] more to the description of the methods, than the methods themselves". I do trust the methods "to be on solid ground" and believe that they are a welcome addition to the literature of diagnostic testing. I recommend the revised manuscript for publication.

**Have the authors made all data and (if applicable) computational code underlying the findings in their manuscript fully available?**

Reviewer #1: Yes

Reviewer #2: Yes

PLOS authors have the option to publish the peer review history of their article (what does this mean?). If published, this will include your full peer review and any attached files.

Reviewer #1: **Yes: **Peter J Diggle

Reviewer #2: No

---

## [Editor Report · Acceptance letter]

19 Apr 2024

PCOMPBIOL-D-23-02039R1 

Combined multiplex panel test results are a poor estimate of disease prevalence without adjustment for test error.

Dear Dr Challen,

I am pleased to inform you that your manuscript has been formally accepted for publication in PLOS Computational Biology. Your manuscript is now with our production department and you will be notified of the publication date in due course.

With kind regards,

Anita Estes
